

# Investigating the spatial representativeness of Antarctic ice cores: A comparison of ice core and radar-derived surface mass balance

Marie G.P. Cavitte[1], Hugues Goosse[1], Kenichi Matsuoka[2], Sarah Wauthy[3], Vikram Goel[4], Rahul Dey[4], Bhanu Pratap[4], Brice Van Liefferinge[5], Thamban Meloth[4], and Jean-Louis Tison[3]

[1]Earth and Life Institute (ELI), Université catholique de Louvain-La-Neuve (UCLouvain), Louvain-la-Neuve, Belgium
[2]Norwegian Polar Institute, Tromsø, Norway
[3]Laboratoire de Glaciologie, Université libre de Bruxelles (ULB), Brussels, Belgium
[4]National Centre for Polar and Ocean Research (NCPOR), Ministry of Earth Sciences, Vasco-da-Gama, Goa 403804, India
[5]SPF Interieur, Brussels, Belgium

**Correspondence:** Marie Cavitte (marie.cavitte@uclouvain.be)

**Abstract.** Surface mass balance (SMB) over the Antarctic Ice Sheet must be better understood to document current Antarctic contribution to sea-level rise. Field point data using snow stakes and ice cores are often used to evaluate the state of the ice sheet's mass balance as well as to validate SMB derived from regional climate models, which are then used to produce future climate projections. However, spatial representativeness of individual point data remains largely unknown, particularly in the coastal regions of Antarctica with highly variable terrains. Here, we compare ice core data collected at the summit of eight ice rises along the coast of Dronning Maud Land, as well as at the Dome Fuji site, and shallow ice-penetrating radar data over these regions. Shallow radar data has the advantage of being spatially extensive with a temporal resolution that varies between annual and sub-decadal resolution from which we can derive a SMB record over the entire radar survey. This comparison allows us therefore to evaluate the spatial variability of SMB and the spatial representativeness of ice-core derived SMB. We found that ice core mean SMB is very local and the difference with radar-derived SMB increases in a logarithmic-fashion as the surface covered by the radar data increases, with for most ice rises a plateau ∼1-2 km away from the ice crest where there are strong wind-topography interactions, and ∼10 km where the ice shelves begin. The relative uncertainty in measuring SMB also increases rapidly as we move away from the ice core sites. Five of our ice rise sites show a strong spatial representativeness in terms of temporal variability, while the other three sites show it is limited to a surface areas between 20-120 km$^2$. The Dome Fuji site on the other hand shows a small difference between pointwise and area mean SMB, as well as a strong spatial representativeness in terms of temporal variability. We found no simple parameterization that could represent the spatial variability observed at all the sites. However, these data clearly indicate that local spatial SMB variability must be considered when assessing mass balance as well as comparing modeled SMB values to point field data.

## 1   Introduction

Quantifying current mass balance and predicting its future evolution is key to constraining the present and future rate and timing of sea level rise due to Antarctic freshwater input. Mass balance is the result of the net equilibrium between surface mass



balance (the net accumulation of snow) at the surface of the ice sheet and the dynamic losses at the grounding line (Lenaerts et al., 2019). The thermodynamics of the atmosphere indicate that with rising temperatures, there will be rising specific humidity (Clausius, 1850). We should therefore expect increasing snow accumulation over the Antarctic Ice Sheet, which could

potentially offset grounding line dynamic losses, for a while at least. Over the past decades, temperatures have been steadily rising, with a large spatial variability over the globe (Arias et al., 2021). To see if snow accumulation is increasing as a result over Antarctica, we can look at direct measurements of accumulation such as firn cores, ice cores and stake measurements, each with their own measurement uncertainties. Firn and ice cores (which we refer to as ice cores from here on out) provide the highest temporal resolution records. The Antarctic coastal region is getting most of the accumulation due to source prox-

imity and low elevation (e.g., Lenaerts et al., 2019). However, when examining the ice core records from this region, they all have very different trends (Medley and Thomas, 2019). A lot of temporal variability is likely introduced by atmospheric rivers (Maclennan et al., 2022), with a high impact along the coastal areas of East Antarctica. Ice cores provide a record of SMB at the seasonal to annual temporal resolution in the shallowest part of the ice column (Fudge et al., 2016; Philippe et al., 2016; Hoffmann et al., 2022). Resolution decreases with depth as snow compacts and lateral ice flow augments the layer thinning. Ice

cores or point measurements can give a high temporal resolution for SMB but their spatial sampling is limited and their surface footprint is quite small. Moreover, wind-driven processes active at the surface can redistribute deposited snow (Frezzotti et al., 2004, 2007; Lenaerts et al., 2019; Kausch et al., 2020; Wever et al., 2022). Measuring SMB in one point (i.e. ice cores) could therefore sample an accumulation record that displays locally higher or lower SMB rates than over the wider region because of the selected location. Wever et al. (2022) have shown that high wind speeds can re-mobilize snow and produce depositional

patterns that are spatially variable on the scale of a few meters. Kausch et al. (2020) have also shown how the interaction of coastal domes' orography and surface winds can create meter to kilometer scale SMB spatial heterogeneity. In addition, it is not simple case of averaging many records in close proximity to reduce noise in ice core records (Cavitte et al., 2020; Münch et al., 2021). To predict future mass balance changes, we rely on modelling simulations. Model simulations are validated using instrument data and paleo data, and in particular ice core data for SMB model outputs. However, observations and simulations

do not always agree (Agosta et al., 2019; Goel et al., 2022; Pratap et al., 2022). The models with the highest spatial resolution to be compared to point measurements are regional climate models which also have specific adaptations of their physics for the polar regions (Van Wessem et al., 2018; Fettweis et al., 2013). Regional climate models have smaller grid cells than global climate models but remain at best 5.5 x 5.5 km in size (Van Wessem et al., 2018). The discrepancies between model results and ice core measurements might be partly due to the difference in their spatial representativeness.

Radar data can be used to interrogate ice cores' spatial representativeness. By connecting radar internal reflecting horizons (IRHs) to ice core sites, SMB rates can be derived over the entirety of a traced radar survey using the ice core age-depth timescale (Verfaillie et al., 2012; Cavitte et al., 2016; Le Meur et al., 2018). Radar-derived absolute SMB are therefore not independent of the ice core SMB, but spatial variations in the SMB can be measured between the ice core site and the wider surveyed area. Ground-based radar surveys are often relatively dense over areas of a few tens of km$^2$, so they provide SMB

constraints at a spatial scale comparable to the models. This raises the new opportunity that instead of comparing ice core point



observations to model simulations, we could compare the radar-derived SMB signal averaged spatially over a model's grid size.

In this study, we compile various radar-derived SMB datasets and example their spatial variability referenced co-located ice core SMB. We compare ice core pointwise measurements to wider spatial averages obtained from subsets of the radar surveys, to be representative of a wider model grid cell. The nine sites studied in this work were chosen because of their data availability and the grid-like design of the surveys which sample SMB over varied surface topography more homogeneously. Eight of these sites are ice rises along the Dronning Maud Land coast where radar stratigraphy has a multi-annual vertical resolution. The radar surveys have a grid-like design that samples in all directions over the ice rises. These eight sites can thus be easily compared. The ninth site is the Dome Fuji plateau region which has a low accumulation rate and a lack of surface topography as opposed to the coastal sites. First, we describe how we derive comparable SMB records for the ice cores and the radar surveys. Second, we examine the spatial differences in SMB between the two, and briefly discuss the meteorological conditions that could explain the patterns observed. Third, we grid the radar-derived SMB to look at the systematic differences in mean and temporal variability of the two SMB records as a function of grid cell size. The ultimate goal of this study is to discuss the implications of these differences for interpreting point measurements of SMB. We provide a detailed assessment of the uncertainties associated with each proxy, and place the SMB differences observed in the context of these uncertainties. In addition, we publish our ice core-radar SMB data, an important first step in the future evaluation of modelled SMB, at (doi will be added upon publication) which will be kept up-to-date with future developments.

## 2   Geographical setting of the survey sites

The coastal sites consist of eight ice rises, located along the Dronning Maud Land coast, stretching from Blåskimen Island ice rise around 3°W to Derwael ice rise around 26°E (Fig 1). Ice rises can be surrounded partially/fully by ice shelves or partially attached to the ice sheet (Matsuoka et al., 2015). Four of our ice rises are in the former setting (Fig. 1, BI, KM, KC and De ice rises) while the others are attached to the ice sheet through a topographic saddle ((Fig. 1, Dj, Le, Ha and Lo ice rises). The ice rise geometries are very variable from ice rise to ice rise, such as Blåskimen Island ice rise showing good radial symmetry while Hammarryggen shows a triple divide and an elongated saddle area connecting it to the main ice sheet. Ice flows radially away from the ice rise summits, with near-to-null ice flow at their crests and close to 10 meters per year near the grounding line, where they meet the ice shelf which flows at speeds between 80 and 800 meters per year (Rignot et al., 2017). Because of their coastal location, the ice rises are the first topographical barrier to incoming synoptic systems and so have high accumulation rates, around tens to hundreds of centimeters of snow per year (Lenaerts et al., 2014; Dalaiden et al., 2020). On average, synoptic marine air intrusions arrive from the northeast along the Dronning Maud Land coast, bringing warm moist air inland (Gorodetskaya et al., 2013; Lenaerts et al., 2014). On the other hand, katabatic winds bringing cold dry air flow down the ice sheet from a southeast mean direction, which combined with the marine air intrusions, results in winds that predominantly blow from east-to-west. As a result, the accumulation pattern on the coastal ice rises is typically high accumulation on their eastern (windward) side and low accumulation on their western (leeward) side (e.g. Goel et al., 2020).

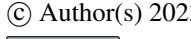



**Table 1.** Radar system characteristics and key references for all radar surveys in this study. See Fig. 1 for site name abbreviations.

| Site | Center radiowave frequency [MHz] | Digitizing step [ns] | Key references |
|---|---|---|---|
| BI | 400 | 0.2930 | Goel et al. (2017b); Vega et al. (2016) |
| KM | 400 | 0.2500 | Goel et al. (2022) |
| KC | 400 | 0.2500 | Goel et al. (2022) |
| Dj | 250 | 0.4000 | Pratap et al. (2022); Dey et al. (2023) |
| Le | 250 | 0.4000 | Pratap et al. (2022) |
| Ha | 400 | 0.2724 | Cavitte et al. (2022), Wauthy et al., submitted |
| Lo | 400 | 0.2724 | Cavitte et al. (2022), Wauthy et al., submitted |
| De | 400 | 0.2931 | Cavitte et al. (2022); Philippe et al. (2016) |
| DF | 5000 | 0.2500 | Van Liefferinge et al. (2021a); Fujita et al. (2021) |

The interior plateau site is centered over the Dome Fuji area (Fig. 1). Dome Fuji is a very wide topographic dome located
along the continental ice divide, where ice flow is practically null and accumulation rates are on the order of $\sim$2.5 cm w.e.
yr$^{-1}$ (Van Liefferinge et al., 2021a). Katabatic winds are practically nonexistent as Dome Fuji is the topographic summit.
Accumulation falls 40 % of the time in the form of diamond dust and 60 % of the time through synoptic precipitation events
(Dittmann et al., 2016).

## 3  Radar data collection over the survey sites

The radar surveys have been collected using ground-based radar systems. For the coastal ice rises, data was collected using
commercial ground-penetrating radar systems (VHF, pulsed), pulled behind a snowmobile. Along-track data density is on
average $\sim$1 m while across-track data density varies a lot per survey design and site. Some surveys are dense grids (e.g.
Leningradkollen ice rise, Fig. 1), while others are radially extending from the center where the ice core site is located (e.g.
Lokkeryggen ice rise, Fig. 1). Table 1 shows the characteristics of each radar system. The digitizing step is the fast-time pixel
size determined by the ratio between the two-way travel time range and the number of sample points. The interior Dome Fuji
site consists of a dense ground-based radar survey collected during the 2018–2019 field season (Van Liefferinge et al., 2021a).
The radar data were collected every 0.5-1 m along transects and between 1-2 km across transects in a gridded survey. The
closest radar transect passes within 79 meters of the NDFN ice core, used for dating the radar isochrones. The radar system
is an ultrawide-band FM-CW radar (UHF, Taylor et al., 2019), operating over 2–8 GHz with a 6 GHz bandwidth, giving a
digitizing step of 0.25 ns (after applying a Hanning time-domain window, CReSIS, 2016), corresponding to $\sim$5 cm in snow.
Radar system characteristics used at Dome Fuji are also given in Table 1.





**Figure 1.** Basemaps of the individual study sites which comprise the eight coastal ice rises and the interior dome. Ice cores and radar surveys used in this study are highlighted as yellow stars and black lines, respectively. **(a)** General view of all sites with the RAMP RADARSAT mosaic (Jezek et al., 2013) as background. **(b)-(h)** Zoomed-in view of the individual radar surveys and co-located ice cores. The sites' initials used in all other figures are provided on each panel: BI=Blåskimen Island; KM=Kupol Moskovskij; KC=Kupol Ciolkovskogo; Dj=Djupranen; Le=Leningradkollen; Ha=Hamarryggen; Lo=Lokkeryggen; De=Derwael; DF=Dome Fuji. Background is the Reference Elevation Model of Antarctica (REMA) elevation (units are meters and referenced to the WGS84 ellipsoid), contour lines are shown at 100 m intervals (Howat et al., 2022). Note that the REMA elevation color bounds are the same for panels (b)-(g), while panel (h) has its own bounds. This figure was prepared with Quantarctica (Matsuoka et al., 2021).





## 4 Methods

Methods used for this work are adapted from the Cavitte et al. (2022) study. We describe them here below more succinctly. We emphasize that to compare SMB measured in the ice cores to that derived from the radar data directly, we need to make sure that the SMB record from the ice core and from the radar are at the same temporal resolution, which is why the ice core SMB is calculated per pair of radar IRH depths.

### 4.1 Ice core SMB

We use the published ice core data, specifically the age, depth and density measurements (Table 1 summarizes the key source data). Using the depth-density raw measurements at each ice core site, we calculate the best-fit depth-density profile between the surface and the deepest radar IRH depth considered, as done previously in Cavitte et al. (2022) (Supplementary Fig. S1). For the coastal sites, we use the exponential function of depth versus density by Hubbard et al. (2013), developed for the Derwael ice core, and adjust it for each ice core site, so that the highest $R^2$ value is obtained for the fit between the raw ice core density measurements and the exponential curve ($R^2$ vary from 0.63 at Le to 0.99 at Lo and Ha, see Supplementary material S1). Note that applying a Herron-Langway depth-density fit (Herron and Langway, 1980), as applied for the radar-derived data gives the same $R^2$ values (within $\pm 0.02$) of the $R^2$ exponential fits. For the Dome Fuji region, we use the linear best-fit outlined by Van Liefferinge et al. (2021a) measured across four shallow cores collected down to 20 m depth in the Dome Fuji area. The best-fit equations obtained at each site are provided in Supplementary material S1 as well as the profiles (Fig. S1). Next, we integrate the best-fit density profile with respect to depth to obtain the cumulative mass as a function of depth in each ice core. Equations linking cumulative mass and depth are also provided for each site in Supplementary material S1. We calculate cumulative mass between the surface and the depth of each IRH, $CM_{IRH_n}$. This allows us to compare the radar and ice core SMB records easily. The mean SMB for each time interval contained between successive pairs of IRHs at the ice core location (where the density profile is known) is obtained by dividing the difference between the deepest and the shallowest $CM_{IRH}$ of successive IRHs by the time interval contained between the pairs of dated IRHs. We note that we use the updated Dey et al. (2023) chronology for the Djupranen ice core.

### 4.2 Radar-derived SMB

#### 4.2.1 Calculating SMB from the IRH data

We use the published IRH data sets at each site to derive a radar SMB record over each region (Table 2 provides a summary of the radar IRH data sets used). Many studies assume horizontal homogeneity of the surface density, often done because only one surface density is measured during a radar campaign, that of the central ice core. However, we know that surface density is highly variable, especially in coastal areas. Wever et al. (2022) have shown that snow events can exhibit up to 150 kg m$^{-3}$ differences in density at the Hammarryggen ice rise. Matsuoka et al. (2015) have shown that there is a 35 % spatial density variation over three ice rises in the Fimbul ice shelf, while Goel et al. (2022) describe a surface density variation of $\pm 2.5$ %





over Blåskimen Island, ±7 % over Kupol Moskovskij and ±2.5 % over Kupol Ciolkovskogo. Therefore, we calculate a density profile at every radar data point, based on fitting the IRH geometries and matching the IRH ages, which allows us to calculate a

SMB history. The first step is to date the IRHs. The ice core best-fit density profile described above is used to convert the IRH two-way travel time (twtt) to depth at the point of closest distance to the ice core site, where we have a measured density profile. Each IRH depth obtained can then be matched to an ice core depth, and each IRH can therefore be dated by linear interpolation of the ice core age-depth timescale. IRHs have different spatial extents depending on their brightness and continuity and so the point closest to the ice core site might vary with the depth of the IRH considered. This is why we report a range of shortest

distances between the IRHs and the ice core sites for some locations in Table 2. Note that the Derwael ice rise site is an exception: the IRH age is the mean age over three locations intersecting the Raymond arch (Cavitte et al., 2022).

The next step is to convert the IRH ttwts and ages into a SMB history, for every radar data point at each site. As stated earlier, density varies a lot in the near-surface. Therefore we use an adaptation of the Medley et al. (2013) algorithm that allows us to invert the IRH ages and twtts to obtain a long-term SMB record with depth (SMB calculated between the surface and

each IRH) at each radar data point. Using the Herron and Langway (Herron and Langway, 1980) depth-density profile, and using the RACMO2.3p2 (Van Wessem et al., 2018) mean surface temperature and mean SMB over 1979-2016 for each site, and the local ice core's surface density as initial guesses, we obtain a density profile which is then converted into a cumulated mass from the surface. Combined with the IRH age information, a first long-term SMB record is obtained. This long-term SMB record is compared to the initial guess SMB and as long as the difference obtained between the prior SMB record and

the resulting SMB record is larger than 0.1 mm w.e. a$^{-1}$ for all time intervals, we iterate using the previous long-term SMB history as an initial guess. Once the difference in long-term SMB history between two iterations is within 0.01 mm w.e. a$^{-1}$ for all IRH pairs (which usually takes ∼4-5 iterations), we consider that we have obtained the best estimate of the long-term SMB history. We can then obtain the mean SMB between pairs of IRHs (i.e. what we refer to as the radar-derived SMB) by dividing the long-term SMB history by the time interval contained between the pairs of IRH, for each radar data point.

In summary, we use the published IRH twtt and the co-located ice core chronologies, to obtain best-fit density profiles, allowing us to calculate a SMB record at each radar data point. This ensures that the SMB rates are calculated from the radar data consistently across all the sites studied. Note that this is why the ages of the IRHs and the SMB history reconstructed from the radar differs slightly from previously published studies. We refer readers to Cavitte et al. (2022) for more details about our method.

We would like to highlight that the ages of the IRHs over the Lokkeryggen and Hammarryggen ice rises have been updated from those provided in Cavitte et al. (2022), which were based on preliminary chronologies (Wauthy et al., submitted). In addition, note that for the Djupranen site, we use the IRH twtts as given in the Pratap et al. (2022) and the recent chronology update published by Dey et al. (2023), while based on comparison of mass balance estimates though different approaches (Goel et al., 2022), we highlight that it is suspected that the age chronology of the Kupol Moskovskij ice core might be in error.



**Table 2.** IRH data set sources and main characteristics. Same abbreviations as in Fig. 1.

| Site(#[*]) | IRH data set reference | Depth[**] range[m] | Age [yr] | Mean age span [yr] | Closest distance to ice core site [m] |
|---|---|---|---|---|---|
| BI(4) | Goel et al. (2017b, a) | 3.9-14.7 | 2010,2009,2005,2001 | 3.1 | 148 |
| KM(7) | Goel et al. (2017b, a) | 3.2-18.0 | 2011,2009,2006,2003,2002,2000,1998 | 2.2 | 98 |
| KC(4) | Goel et al. (2017b, a) | 3.1-13.0 | 2004,1995,1989,1981 | 7.7 | 889 |
| Dj(6) | Pratap et al. (2022, 2021) | 3.3-13.5 | 2012,2011,2006,1997,1991,1989 | 4.8 | 318-325 |
| Le(5) | Pratap et al. (2022, 2021) | 1.6-9.7 | 2012,2007,2002,1997,1988 | 5.8 | 91 |
| Ha[†](6) | Cavitte et al. (2022); Cavitte (2022) | 6.5-35.9 | 2011,2007,1993,1980,1970,1956 | 10.3 | 27-42 |
| Lo[†](7) | Cavitte et al. (2022); Cavitte (2022) | 15.0-34.0 | 2000,1995,1994,1990,1984,1981,1977 | 5.7 | 16-26 |
| De(6) | Cavitte et al. (2022); Cavitte (2022) | 11.2-38.7 | 2003,1993,1988,1985,1980,1972 | 6.6 | *** |
| DF(3) | Van Liefferinge et al. (2021a, b) | 5.4-14.4 | 1940,1924,1772 | 82.4 | 794 |

[*] Brackets indicate the number of isochrones in the radar data set.

[**] Depths are measured at the point of closest approach to the ice core site, except for Derwael where depths are the average over the three equivalent locations, as described in Section 3.1.

[***] The closest distance does not apply to the Derwael ice rise, as described in Section 3.1.

[†] Note that IRH ages for the Lo and Ha sites have been updated from those provided in Cavitte et al. (2022) which were based on preliminary ice cores chronologies.

## 4.2.2 Gridding SMB

To quantify the spatial representativeness of the individual point SMB measurements, i.e. the ice core measured SMB, we compare them to the radar-derived SMB obtained over a larger surface for each region. We minimize the sampling bias of the radar data, which is orders of magnitude denser along-transect than across-transect direction (meter versus kilometer, respectively), by re-sampling the radar survey data onto a regular square grid, as described in Cavitte et al. (2022) for three square grid cell sizes: 50x50, 100x100 and 250x250 m. This smooths out the spatial heterogeneity of the individual radar survey designs. The SMB record of each grid cell is the average of the SMB records for all the radar data points that fall within that grid cell. We compared the mean SMB obtained with and without gridding for the entire survey region, and the difference is insignificant with respect to the SMB uncertainties for all sites in terms of differences in mean SMB, but significant for two ice rise sites in terms of temporal variability (9 % and 11 % increases in temporal variability with the gridded product, see Section 5.1 for the uncertainty quantification).

The compiled data sets, namely the gridded radar-derived SMB data and the ice core SMB data derived at the radar temporal resolution are available at (doi will be added upon publication).





## 5   Results

### 5.1   Error analysis

#### 5.1.1   Radar-derived SMB uncertainty

For the radar, the sources of SMB uncertainty are two-fold, related to errors in the densities and the measured IRH twtt thickness that results from the radar system used to collect the radar data. We quantify the density error as the standard deviation across all the density profiles per radar survey (Supplementary Fig. S2), following the approach by Medley et al. (2013). This density error affects the SMB calculated by modifying the cumulative mass between IRH pairs. The impact of this mass error is

assessed by calculating the SMB between IRH pairs, once using the mean density profile (mean of all density profiles per radar survey), and again using the mean density profile from which we add or subtract the density error. The difference obtained between the SMB derived using the mean densities and the SMB derived using the modified densities (density±std dev of the density) gives us the contribution to the total SMB error as a result of this mass error. This density-related mass error is estimated to have a range between 0 % and 4.55 % across all the radar survey sites (Fig. 2). The density error also affects the

calculated SMB by impacting the conversion of the IRHs' twtt to depth. The impact of this density-related depth error on the SMB history calculated can be assessed in a similar way as above, but this time by evaluating the twtt-to-depth conversion using the mean density profile or modified by the density error. This density-related depth error is estimated to have an impact up to 2.80 % on the calculated radar-derived SMB across all study sites (Fig. 2). Finally, we assign each IRH a twtt-thickness uncertainty equal to four times the digitization step for each radar data set (Table 1), as we observe the traced radar IRHs to

be typically ~4 pixels wide in the twtt domain. This measured IRH twtt-thickness uncertainty affects the calculated SMB by impacting the measured twtt of the radar IRHs. We calculate the SMB between IRHs pairs by using the IRH twtts modified by ±2 times the digitization step, which we compare to the calculated SMB without IRH twtt modification. The resulting SMB error is estimated to be between 0.40 % and 6.94 % of the radar-derived SMB across all study sites (Fig. 2). We highlight that the relative significance between the assigned digitizing error (4 pixels) and the twtt of the IRH (i.e. total number of pixels)

has a strong influence on the resulting SMB error magnitude. For example, the Le and Dj sites have the same digitization step, however, because the IRHs traced over Le have shallower depths than those traced over the Dj site, the relative SMB error is larger at Le than at Dj. This is clearly visible in Fig. 2.

We combine these three errors as a root-mean-squared error, resulting in a total radar-derived SMB uncertainty across all sites that varies between a minimum of 0.45 % of the mean radar-derived SMB for the interval between the surface and the

shallowest IRH at Lo, and a maximum of 6.94 % of the mean radar-derived SMB for the interval between the two deepest IRHs at each site at Le (Fig. 2). Because SMB is evaluated as a cumulative mass from the surface, SMB uncertainties increase with depth (and age) of the IRHs. If we average the total radar-derived SMB uncertainty across all sites, we get a mean radar-derived SMB uncertainty of 1.87 % near the surface and 3.54 % for the deepest IRHs. Detailed site-by-site SMB uncertainties are provided in supplementary material S3, Table S1.





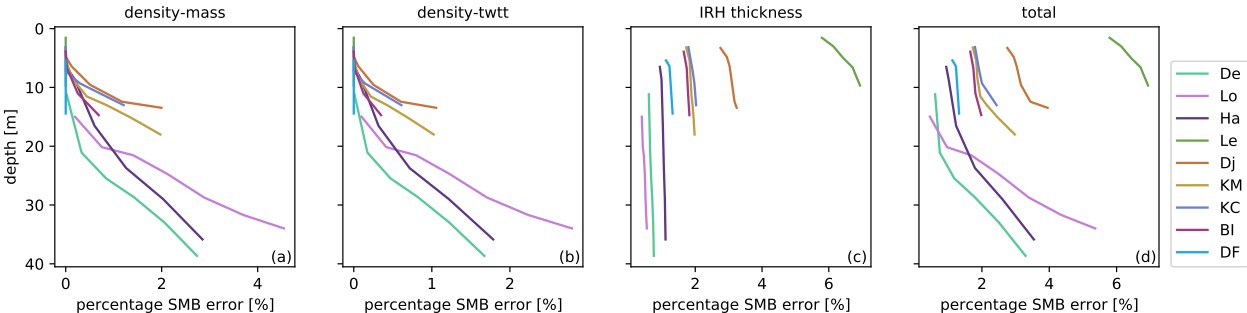

**Figure 2.** Radar-derived SMB uncertainty due to (a) density-related mass error, (b) density-related twtt error, (c) measured IRH twtt thickness, (d) combined SMB uncertainty. Each studied region is portrayed by a different color. SMB uncertainty increases with depth as SMB is calculated with respect to a cumulative mass between time markers. Note that the density-related mass error and the density-related twtt error is the same for Le and Dj and so the two curves overlay each other. Abbreviations are the same as in Fig. 1.

### 5.1.2 Ice core SMB uncertainty

For the ice core, the sources of SMB uncertainty are two-fold, related to the error made in estimating a best-fit from the raw density measurements and to the accuracy in measuring the individual ice core section lengths during drilling operations. Since we are comparing ice core and radar-derived SMB, the age scale applied is identical and we can ignore the ice core dating uncertainty. We quantify the density-related error by calculating the standard deviation error between the ice core raw measured densities and the exponential fit of the densities, at each site. We note that this calculated density error matches that of the ice core density measurement errors where reported (Hubbard et al., 2013; Pratap et al., 2022; Dey et al., 2023). We then evaluate the impact of this density error on the calculation of the ice core SMB. The total SMB error is determined as the difference between the SMB obtained using the best-fit density profile and the SMB obtained using the best-fit density profile ± the density error. This is assessed by calculating the cumulative mass to each ice core depth marker, using the three density profiles determined. This cumulative mass is then interpolated to obtain the cumulative mass to each IRH depth, which we divide by the time interval contained between IRH pairs as before. This density-related mass error is estimated between 1.3 % and 8.5 % of the ice core SMB across all survey sites, except for the NDFN ice core where this error varies between 13.7-16 % of the ice core SMB (Fig. 3). This larger relative error for NDFN is due to the fact that the absolute snow accumulation rates are a factor of 100 smaller than at the coastal sites, and so a small mass error corresponds to a large relative SMB error. Note that the exponential fit is often worse near the surface (Supplementary Fig. S1) and so the SMB uncertainty related to this density error is largest near the surface and decreases with depth, for all regions studied. Fig. 3, panel a, shows this inverted relationship well. Second, we evaluate the depth error due to core length measurement error. This depth error is rarely reported in publications in general because of its negligible impact versus errors made in the dating of the ice. Since it is not reported for the ice core data sets used, we make the assumption that a 1 mm measurement error is made for every 0.5 m of ice core drilled, for all ice core sites considered in this study, which accumulates with depth. A depth error is then assigned to each IRH



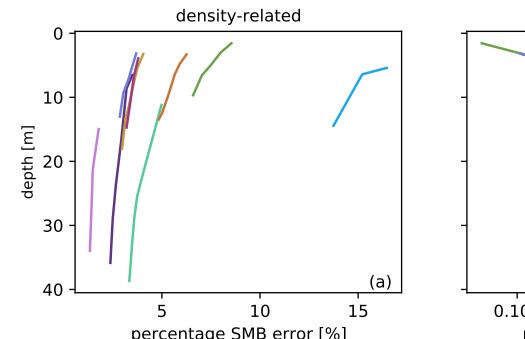 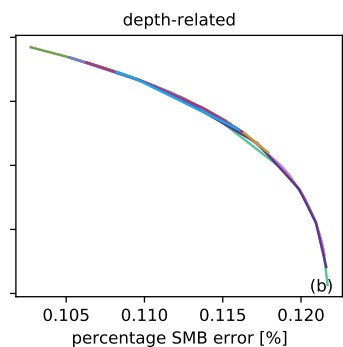 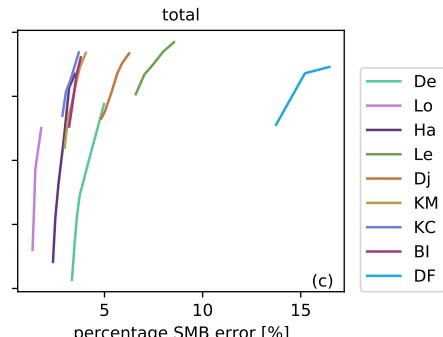

**Figure 3.** Ice core SMB uncertainty due to (a) density-related SMB error, (b) depth-related SMB error, (c) combined total SMB uncertainty, each study site is portrayed by a different color. SMB uncertainty decreases with depth due to the improved fit of the exponential density profile away from the surface. Abbreviations are the same as in Fig. 1.

depth and we calculate the SMB using the exponential functions described in Section 3.1, once for the IRH depth and then for the IRH depth ± half of the IRH-assigned depth error. This results in an ice core SMB error around 0.1 %, including at NDFN. This depth-related SMB error is ten times smaller than then density-related SMB error and can therefore be ignored. The total ice core SMB error is therefore equal to the density-related error (Fig. 3).

240  We observe that the ice core SMB uncertainties are largest at the surface due to the larger number of measurement outliers near the surface and where the exponential profile is less adapted. The SMB error is largest for the NDFN core as the misfit between the linear profile and the measurement points is strongest, enhanced by the low SMB absolute values. The SMB error decreases with depth with increasing fit of the exponential profile to the measurements (Supplementary material S1, Fig. S1). Detailed site-by-site SMB uncertainties are provided in Supplementary material S3, Table S2.

## 5.2 Spatial patterns of SMB

### 5.2.1 Coastal ice rises

All ice rise sites show a clear east-west spatial pattern with higher SMB values on the eastward side and lower SMB values on the westward side of the ice rise crests. With the dominating east-to-west wind patterns, the eastward side of the ice rises corresponds to the windward side and the westward side to the leeward side. The ratio of the mean windward SMB to the mean leeward SMB varies between 1.1 at Lokkeryggen and 2.3 at Kupol Ciolkovskogo, with the exception of Djupranen whose ratio is 1, perhaps linked to the survey design which is uniformly split in sampling the windward and leeward slopes of the ice rise (Fig. D3). Blåskimen Island ice rise is shown as an example in Fig. 4 and all the other sites can be found in Supplementary material S4. We note that Leningradkollen ice rise shows a less clear east-west pattern. Instead, it shows higher values on the southern side of the east-west ridge and in the saddle between the seaward and the landward ridges, already described in Pratap et al. (2022). SMB values vary between 10 and 150 cm w.e. yr$^{-1}$ across all coastal ice rise sites, and the spatial patterns of



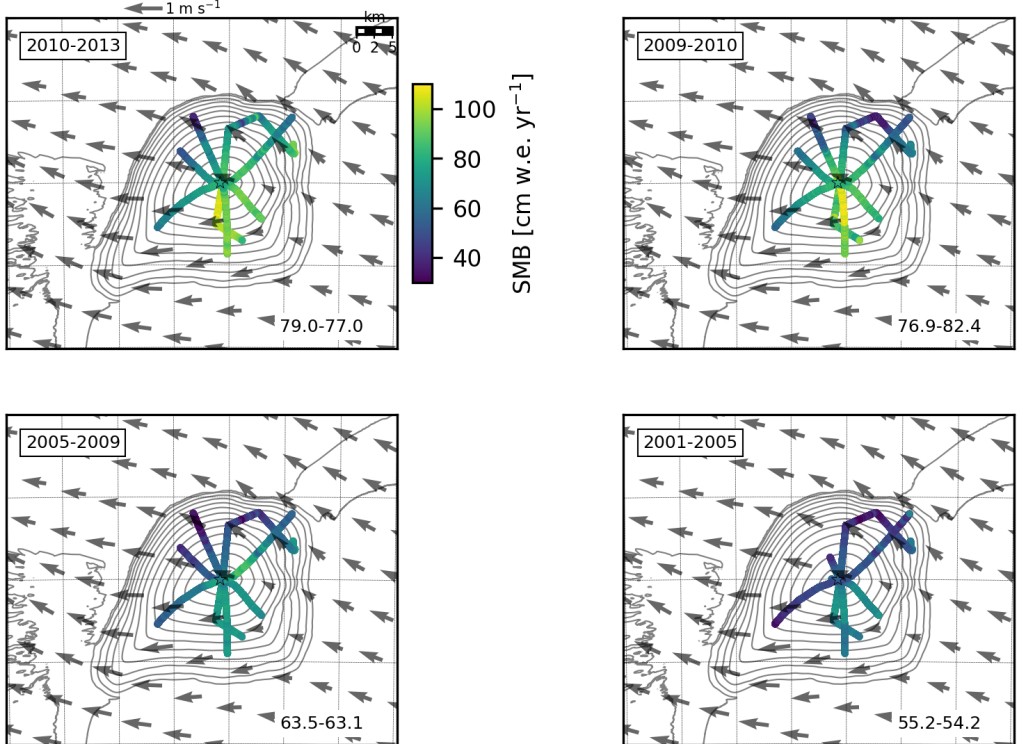

**Figure 4.** Spatial distribution of SMB through time at Blåskimen Island. Each panel represents a different time interval, going from most recent at the top left, to the oldest at the bottom right, the time intervals are provided in the top left corner of each frame. Contours are REMA v2.0 elevation contours, with a 30 m interval, gray arrows show the mean wind direction (RACMO2.3 5.5 km simulations over 1979–2017, Lenaerts et al. (2017); Van Wessem et al. (2018)). The wind magnitude scale is shown on top of the first panel. Numbers in lower right corner of each panel are radar area average SMB followed by ice core SMB in cm w.e. yr$^{-1}$.

SMB remain relatively stationary over the different time intervals. Note that the absolute values of SMB at Lo and Ha are significantly different from the Cavitte et al. (2022) study as a result of the updated ice core chronologies as mentioned above. We do not go into more detail on the spatio-temporal patterns of SMB at each site as these are described at length in the source data set publications (Pratap et al., 2022; Goel et al., 2017b, 2022; Cavitte et al., 2022).

### 5.2.2 Dome Fuji

The interior Dome Fuji site shows a factor of ten lower SMB rates than over the coastal ice rises, ranging between 1 and 2.5 cm w.e. yr$^{-1}$, with a very heterogeneous SMB spatial pattern -no clear east-west pattern- that is stationary over all three time intervals. Van Liefferinge et al. (2021a) describe more at length the spatial distribution of SMB.



## 5.3 Temporal changes of region-mean SMB

### 5.3.1 Coastal ice rises

From the Blåskimen Island ice rise example in Fig. 4, we can see that as we move away from the ice core location, SMB varies greatly on the kilometer scale. To compare more systematically the ice core SMB measurements to the SMB derived over a wider area, we calculate the spatial mean of the radar-derived SMB over the entire radar survey for each site, as done previously in Cavitte et al. (2022). Fig. 5 shows the resulting SMB histories for each site examined. The first thing to note is that in all cases, SMB from the ice cores is smaller than the regional mean, except to a degree at the Blåskimen Island site. The mean difference between the local and the regional SMB signals, given as the $\Delta\mu_t$ value at the top of each panel on Fig. 5, varies significantly between sites. At Blåskimen Island, the mean difference is quasi null, while it reaches up to $\sim$24 cm w.e. yr$^{-1}$ at the Djupranen site, corresponding to 85 % of the Djupranen ice core's SMB temporal mean.

Secondly, it is interesting to note that the SMB temporal evolution of the point measurement and of the area average are very similar for some sites (e.g. this particularly the case for the Hammarryggen, Lokkeryggen and Derwael sites) and less so for other sites (e.g. Kupol Moskovskij, Djupranen and Leningrakollen sites, Fig. 5). Combining ice core and radar measurements allows us to estimate the uncertainty in measuring SMB locally $\Delta obs_t$ for the surface area sampled by the radar survey by quantifying the amplitude of the difference between the point measurement and the area average at each site (after removing the mean SMB bias between the two proxies). In practice, we define $\Delta obs_t$ as the standard deviation of the difference between the ice core and the radar-derived SMB anomalies (i.e. SMB record minus its temporal mean) at each site. $\Delta obs_t$ is given at the top of each panel on Fig. 5 (equations in Supplementary material S5). We then normalize $\Delta obs_t$ by the ice core SMB temporal variability, differing from the method used in Cavitte et al. (2022) where $\Delta obs_t$ (%) was given relative to the ice core mean SMB. $\Delta obs_t$ (%) is therefore the relative uncertainty in the SMB temporal variability, i.e. the ratio of the uncertainty in the SMB temporal variability to the SMB temporal variability. $\Delta obs_t$ (%) varies greatly from site to site, reaching from a maximum of 180 % over the Kupol Moskovskij ice rise down to 26 % at Blåskimen Island. In the definition of the relative uncertainty in the SMB temporal variability $\Delta obs_t$ (%) given here, a value of 100 % or above implies that the signal uncertainty is equivalent to or larger than the local signal strength. It corresponds to four ice rises sites in our sample: Kupol Moskovskij, Djupranen, Leningradkollen and Lokkeryggen where the SMB signal strength is weaker than the regional noise brought in from the spatial variability of the SMB.

### 5.3.2 Dome Fuji

The Dome Fuji site is a little different than the coastal ice rise sites due to the extremely low accumulation rates observed at this site and the very SMB uncertainties obtained. At Dome Fuji, the point measurement and spatially averaged SMB histories are very similar in terms of trend and temporal mean. The mean difference is quasi null (3 % of the ice core mean SMB) and clearly much smaller than the SMB uncertainties calculated for the ice core SMB record. The Dome Fuji ice core shows a relative uncertainty in the SMB temporal variability $\Delta obs_t$ (%) of 55 % which implies that the local SMB signal strength is above the SMB noise.

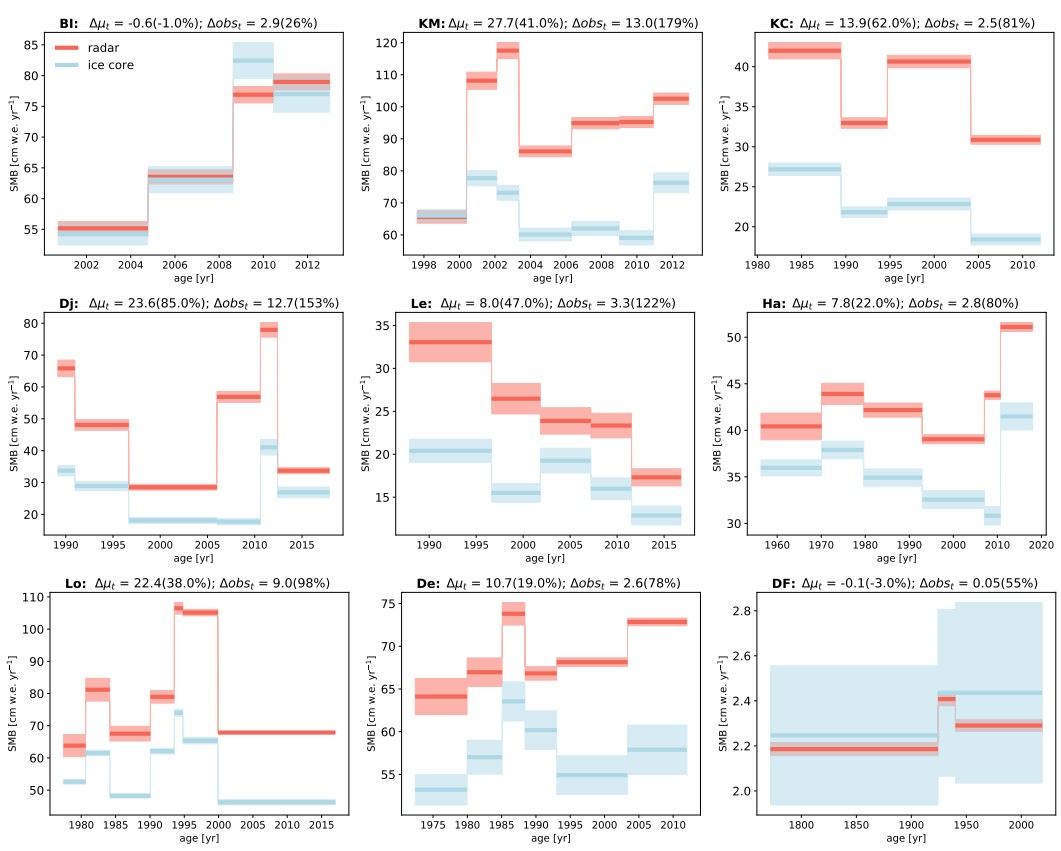

**Figure 5.** Ice core (blue) vs regional (radar spatial average, red) SMB history, with the calculated SMB uncertainties as blue and red bands, respectively (legend is provided on the top left-most panel). Sites are labeled at the top of each panel and are ordered left to right from the western most coastal site to the eastern most coastal site, and the Dome Fuji site is last. On top of each panel, $\Delta\mu_t$ indicates the difference in the mean SMB between the two SMB series in cm w.e. $yr^{-1}$, with difference given as a percentage of the ice core mean SMB in brackets. $\Delta obs_t$ indicates the uncertainty in measuring SMB locally between two SMB series in cm w.e. $yr^{-1}$, with relative uncertainty as a percentage value in brackets (normalized to the standard deviation of the ice core SMB anomaly around the temporal mean).





### 5.4 Spatial representativeness of ice-core-derived SMB

#### 5.4.1 Coastal ice rises

We note that comparing the radar spatially-averaged SMB to the measured ice core SMB in terms of a single mean SMB value
(Fig. 5) is difficult with the varying radar coverage between sites. To quantify the representativeness as a function of distance
from the ice core sites across all study regions, we plot the difference in mean and in temporal variability (calculated as before)
between the ice core SMB and the regional SMB (radar spatial average) for varying surface areas centered on the ice core sites
(Figs. 6 and 7). We use the 50 m gridded radar product, but we note that the results are unchanged if the 100 m and 250 m
grid resolutions are used (see Supplementary material S6 and Fig. S11). We can see that as the surface area over which the
radar spatial average is calculated increases, the mean SMB difference between the point measurement and the area average
increases rapidly too. For the KM, Lo, Ha and De sites, the difference in the mean SMB seems to plateau for a surface area of
$\sim$8 km$^2$ at which point the mean differences reach a maximum between 20 % and 40 % of the ice core mean SMB. At the KC,
Le and Dj sites, the difference in mean SMB continues to grow with increasing surface areas and only stabilizes for a surface
area $\sim$300 km$^2$ and the difference between the ice core SMB and the spatially-averaged SMB reaches between 50 % and 85
% of the ice core mean SMB. At the BI ice rise, the difference in mean SMB between ice core and area average increases with
surface area, but remains low, below 5 % of the ice core mean SMB for all surface areas considered. Regarding the relative
uncertainty in measuring SMB locally $\Delta obs_t$ (%) for the surface area sampled by the radar survey, we see that this uncertainty
increases rapidly for all ice rise sites (except BI), stabilizing below 100 % for the Lo, KC, Ha and De sites while for the Le, Dj
and KM sites, the 100 % threshold is crossed for surface areas varying between 20 and 120km$^2$ and plateaus out at a relative
SMB uncertainty equal to 125-175 % of the temporal variability of the ice core SMB signal. The BI ice rise site stands out
with a very low relative SMB uncertainty $\Delta obs_t$ (%), remaining around 25 % of the ice core SMB signal for all surface areas
considered. This implies that Lo, KC, Ha, DE and particularly BI, the signal strength is higher than the noise introduced by
spatial variability of the SMB signal, and the inverse is true for the Le, Dj and KM sites.

#### 5.4.2 Dome Fuji

At the DF plateau site, the difference in mean SMB between ice core and area average increases with surface area, but remains
low and negative, between -3 % and 0 % of the ice core mean SMB for all surface areas considered, which implies that the ice
core measured SMB is generally higher than that of the area average. In terms of relative uncertainty in measuring SMB locally
$\Delta obs_t$ (%), the plateau DF site shows an opposite trend to the ice rises, with a maximum value when the smallest surface area
is considered ($\sim$200 % of the temporal variability of the ice core SMB signal) and a plateaus out $\sim$100 % for a surface area
$\sim$300 km$^2$.

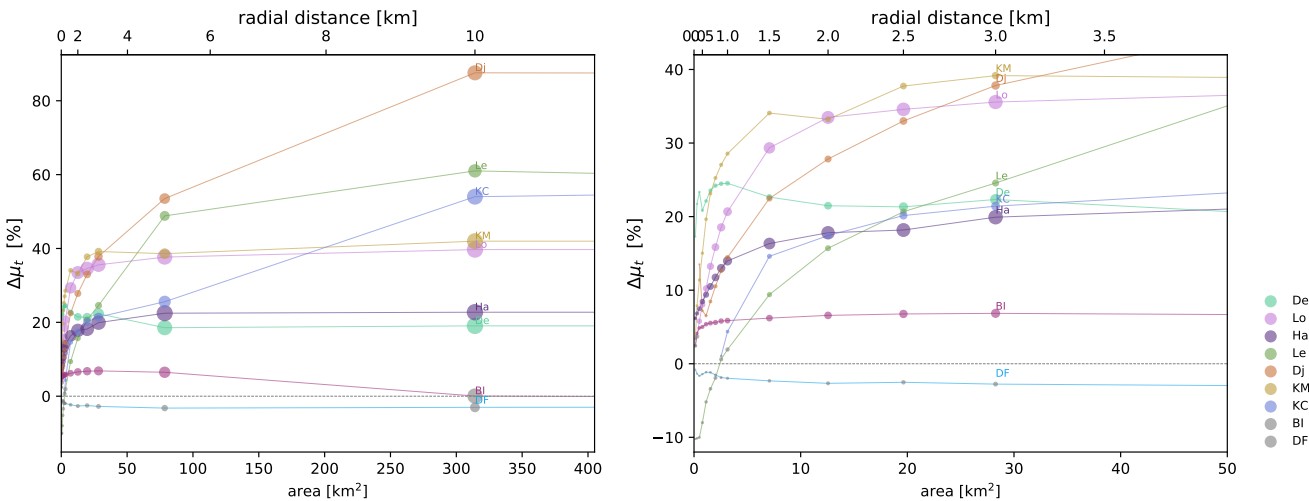

**Figure 6.** Relative difference (%) in mean SMB value between gridded radar-derived SMB (at 50 m spatial resolution) and ice core SMB, as a function of surface area. Right panel focuses on surface areas up to 50 km² around the ice core site. Each site is represented by a different color, gray dots imply that the difference is negligible with respect to the SMB uncertainties. The size of the dots represents the number of grid points within the radial distance. "ic" stands for ice core.

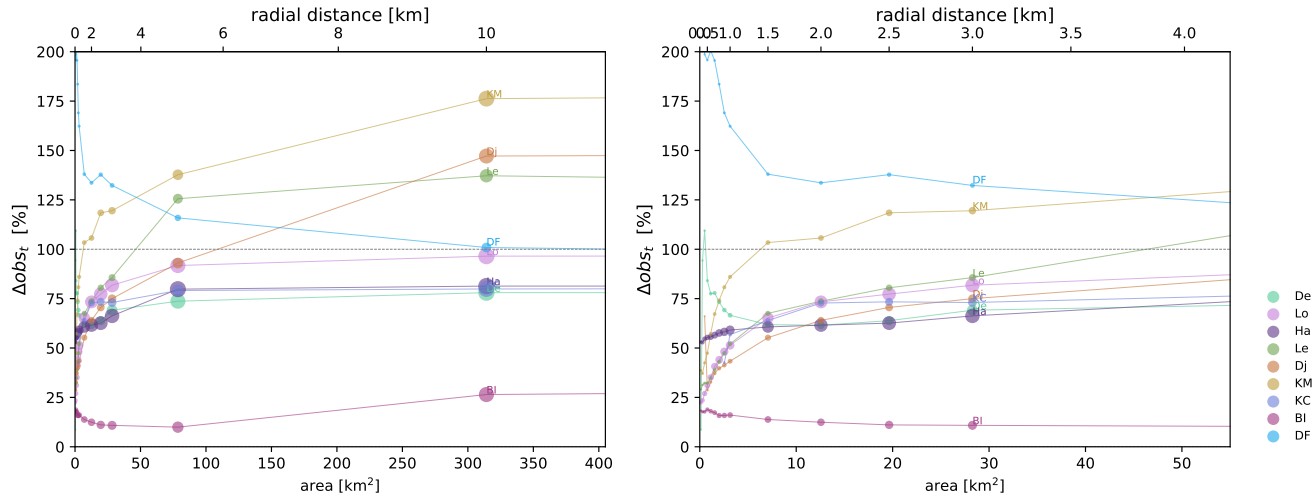

**Figure 7.** Relative uncertainty in measuring SMB locally $\Delta obs_t$ (%). Right panel focuses on surface areas up to 50 km² around the ice core site. Each site is represented by a different color, gray dots imply that the difference is negligible with respect to the SMB uncertainties. The size of the dots represents the number of grid points within the radial distance.





## 6   Discussion

Cavitte et al. (2022) have shown that for the De, Lo and Ha ice rises, the spatial representativeness of the ice cores in terms of mean SMB was limited to a distance ∼200-500 m away from the ice core site. This conclusion was based on the criteria that the difference between the local and the regional signals became smaller than the SMB uncertainties measured. That study showed
therefore that beyond 500 m, the ice core mean value always underestimated the regional mean which could be adjusted based on the regional mean to be representative of a wider region. With the addition of five ice rise sites and an interior plateau site, we observe here that the ice cores' representativeness varies widely from site to site. Considering the difference in mean SMB between a point measurement of SMB and the wider area average, as in Fig. 6, it is clear that these differences cannot be explained through a simple relationship such as distance away from the ice core site. We note that similarity at all ice rise
sites, the mean difference in SMB between a point measurement of SMB and the wider area average increases with the size of the surface area considered, but that the rate at which this difference increases varies from site to site. Since our highest calculated uncertainty is ∼8 % of the SMB signal for both the ice core and the radar, we describe the representativeness of point measurements using the threshold of a 10 % difference in mean SMB between the local and the regional signals, as opposed to the uncertainty value threshold as in Cavitte et al. (2022). This allows us to explicitly recognize that ice cores'
representativeness exists on a continuous spectrum and to ensure that the results are not dependent on the quality of the data (i.e. higher quality radar or ice core could decrease the error and so influence the determined representativeness). The difference in mean SMB increases beyond 10 % of the ice core mean SMB for a surface area that varies between a minimum of ∼0.4 km$^2$ at KM and a maximum of 9.6 km$^2$ at Le. It is interesting that for five of the ice rise sites, the difference in the mean SMB plateaus for a surface around ∼8km$^2$, which corresponds to ∼1-2 km away from the ice rise crests. This is the typical distance over
which a dome and crest SMB pattern is observed over the ice rises (Kausch et al., 2020; Goel et al., 2022) and so this plateau could be linked to the depositional and erosional SMB regimes near the ice rises' summits. The second plateau is reached by the KC, Le and Dj sites for a surface area of ∼300 km$^2$, corresponding to ∼10 km away from the ice rise crests, which is typically the distance where the orographic SMB regime of the ice rises meets the flat ice shelves. For BI and DF, the mean difference remains below 10 % of the ice core mean SMB for all surface areas considered. This suggests that both ice cores
have a good representativeness of the larger region. At the other end of the spectrum, the Dj and Le ice cores record the most local SMB signal with a difference in mean SMB between the point measurement and the regional mean increasing rapidly, up to 90 % and ∼60 %, respectively, of the ice core mean SMB for a surface area of 300 km$^2$. If we consider that regional climate models often have a spatial resolution on the order of 25 km x 25 km (= 625 km$^2$), this has important implications for the use of ice cores as model simulation validations. As previously suggested in Cavitte et al. (2022), an option would be to shift the
mean SMB value of the ice core series based on the assessed difference in mean SMB with a co-located radar-derived SMB series.

Cavitte et al. (2022) had also shown that the SMB temporal variability tends to be mirrored between the ice core measurements and the radar-derived average SMBs for the Ha, Lo and De sites, also visible here on Fig. 5. In this extended study, we estimate the uncertainty in measuring SMB locally $\Delta obs_t$ to represent a certain surface area and show that for most ice rise



sites, this uncertainty increases rapidly as the size of the surface area considered increases. Considering the relative uncertainty

$\Delta obs_t$ (%), normalized to the ice core's SMB temporal variability, five ice rises out of the eight (Lo, KC, Ha, De and BI)

show a $\Delta obs_t$ (%) that remains below 100 % of the ice core temporal variability, which suggests that the ice cores sample a

temporal variability signal that is relatively strong and representative of a relatively large surface area. We note the very low

$\Delta obs_t$ (%) value at the BI ice rise, implying the ice core has the highest reliability in sampling the temporal variability of the

area. For the DF area, the interpretation of $\Delta obs_t$ might be limited by the fact that only three time intervals are available to

calculate the temporal variability of the signal, and that the SMB rates are so low (on the order of a few cm w.e. yr$^{-1}$) that even

small errors in the radar-derived SMB from the iterations or the density best-fit for the ice core SMB will have a very large

impact on the final differences obtained. For the KM, Dj and Le ice rises, the difference increases beyond 100 % of the ice

core temporal variability which implies that they record a more noisy record of SMB temporal variability. 100 % difference in

temporal variability is reached between a surface area of ∼20 km$^2$ for KM, ∼50 km$^2$ for Le and ∼120 km$^2$ for Dj.

It is interesting to note that the Dj and Le ice rises are two adjacent sites, in the Nivlisen ice shelf, where SMB ice core

estimates are on the less representative end in terms of both mean and temporal variability of the SMB of the wider area.

If we, again, take the comparison to a typical regional climate model resolution of 25 km x 25 km, our results suggest that the

ice cores studied here contain the temporal variability of the SMB signal with varying degrees of noise. This spatial variability

of the SMB temporal variability adds noise in the ice core SMB records which cannot be easily corrected by a shift as for

the mean SMB representativeness. We suggest that these estimates of the uncertainty in measuring SMB locally to represent

a specific surface area could be included in the models. This would ensure we take into account the uncertainty linked to the

spatial variability of the SMB regionally, assessed on the difference in temporal variability with co-located radar-derived SMB.

We have attempted to find the controlling factors behind the varying representativeness of the ice cores' mean SMB. Several

factors could explain the spread in mean SMB representativeness across all sites considered: varying absolute snowfall rates,

distance from the coast, wind strength, temporal resolution of the radar isochrones considered, as well as radar survey sampling

of the windward and leeward sides (Goel et al., 2017b; Cavitte et al., 2020; Kausch et al., 2020; Pratap et al., 2022; Goel et al.,

2022). We have made a preliminary analysis of the influence of each factor individually (see Supplementary material S6 and

Fig S12), but no single factor explains the spread observed in mean SMB representativeness. It is most likely a combination of

several factors that can explain the varying representativeness.

We recognize that one limitation of our study is the differing survey design at each site which unequally sample the windward

and leeward sides of the ice rises. It has been shown previously that the windward side of an ice rise receives more accumulation

than the leeward side which is more erosional (Kausch et al., 2020; Goel et al., 2022). A heterogeneous sampling could induce

a radar measurement bias that would result in a bias in the ice core-radar differences calculated. To determine the impact of

the surveys' sampling, we use the 50 m gridded radar product to calculate the SMB mean over the windward and leeward grid

points (east and west side of the ice rises' crests, respectively), which are then averaged together, to compare to the SMB mean

obtained previously (i.e. calculated over the whole radar area at once). We calculated that the heterogeneous sampling of the

ice rises due to the survey designs here induces a bias in the mean SMB with a minimum of 0.3 % of the mean SMB at De and

KM and a maximum of 8 % of the mean SMB at Le. This is falls within the radar-derived SMB uncertainties (Supplementary





Table S1) for all rises except two (Le and KC) and can therefore be ignored. For Le, we do not attempt to correct the bias
due to the difficulty in determining the windward and leeward slopes as a result of the ice rise's geometry with respect to the
dominant wind direction. As for KC, this can be explained due to the larger disparity in radar sampling on the windward versus
leeward sides of the ice rise.

If we look at the high plateau site of DF individually, we note that it shows very small differences in mean SMB for all
surface areas considered, albeit negative (the ice core mean SMB is higher than the area average), varying between -3 % and
-0.8 % of the ice core mean SMB. At this site, the SMB rate is very low, around 1-2.5 cm w.e. $yr^{-1}$, and precipitation falls 40
% of the time in the form of diamond dust (Dittmann et al., 2016). It has been demonstrated that wind-induced erosion and
redeposition has a very large impact on SMB distribution (Frezzotti et al., 2004; Lenaerts et al., 2014, 2019; Kausch et al.,
2020). The DF site is located in the interior, where surface wind speeds are low (Fig. D8), therefore we could suppose the
wind-induced redeposition is very limited at DF. There is significant spatial variability of SMB which varies between 1.5-2.5
cm w.e. $yr^{-1}$ for a single time interval, on the same order of magnitude as the SMB accumulation over the time interval, which
Van Liefferinge et al. (2021a) have shown is linked to fine-scale surface topography.

## 7 Conclusions

We have obtained SMB from radar surveys over eight coastal ice rises and one interior plateau site which we were then able
to compare to the ice core measured SMB for each site. We have shown that we can use the radar-derived SMB to estimate
the spatial representativeness of single ice core SMB records. We conclude that the temporal mean SMB measured in ice cores
is representative of varying surface areas, which does not depend linearly on any one meteorological or topographical factor.
We do highlight that there seems to be link between the spatial variability of the SMB and the controlling factor for the SMB
regime (crest-wind interactions versus shelf-to-ice rise orographic interactions). For a mean difference of 10 % between ice
core and radar-derived SMB, ice core SMB is representative of surfaces between 0.5 $km^2$ and 10 $km^2$, corresponding to radial
distances <1 km away from the ice core site). For our westernmost coastal site (BI) and our interior site (DF) however, the mean
difference between ice core and radar-derived SMB always remains below 10 %. Examining the spatial representativeness of
the ice cores' SMB temporal variability signal, we showed that the relative uncertainty in measuring SMB locally $\Delta obs_t$ (%)
for the surface area sampled by the radar survey increases rapidly as we move away from the ice core sites but five of our sites
remain below the 100 % threshold for all surface areas considered, implying that their representativeness in terms of temporal
variability is relatively strong, while for the other three sites, it is limited to a range between 20-120 $km^2$. Our gridded radar
SMB products, as well as the differences in mean SMB and temporal variability of the SMB signals between the ice cores and
the radar-derived data as a function of distance can be found at (doi will be added upon publication).

All our results tend to indicate that SMB measured in ice cores has a local signature. Although regional climate models
have a relatively high resolution, theirs grids remain very coarse compared to the spatial footprint of ice cores. High resolution
simulations such as those of RACMO5.5 (Lenaerts et al., 2017; Van Wessem et al., 2018) with a grid size of 5.5 x 5.5 km, can
be compared directly to ice core SMB if we correct for representativeness, calculated in this study for each site. According

to our results, the error of representativeness in mean SMB between the ice core and the area mean over an area of 30 km$^2$ varies between $\sim$-3 %-35 % of the ice core mean SMB for the sites we studied here. And for the temporal variability, the
error of representativeness varies between $\sim$15-130 % of the ice core SMB temporal variability, which could be integrated in a model-ice core comparison.

Another interesting exercise would be to use the radar-derived regional means, over any surface area of interest, to compare directly to model outputs. The radar-derived mean SMB and temporal variability could be calculated for any subset of the radar survey, and compared to the model grid cell SMB mean and temporal variability, taking into account the radar-derived SMB
uncertainties. This would be complementary to the comparison to ice cores.

As a next step, it will be interesting to compare our derived radar SMB products at each site to regional climate model SMB simulations, in order to assimilate this new SMB data product in model ensemble simulations. Indeed, data assimilation methods need a well defined data error prior, which includes measurement uncertainties and the error of representativeness, both detailed in our work.

**8 Data availability**

The Lo and Ha ice cores' full resolution chronologies and density data will shortly be made available in Wauthy et al (submitted). The radar-derived SMB (gridded and ungridded), as well as the ice core SMB at the radar-temporal resolution will be made available upon publication at (doi will be added upon publication).

*Author contributions.* MGPC led the data analysis and writing. HG and KM provided support in planning experiments and finalizing uncer-
tainty quantification. VG, RD, TM, SW and JLT provided ice core age-depth-density data and support regarding ice core data analysis. VG, BP and BVL provided radar isochrone data and support. BVL provided statistical analysis for uncertainty analysis. All co-authors contributed to discussions on this work. MGPC prepared the manuscript with contributions from all co-authors.

*Competing interests.* The authors declare that they have no conflict of interest.

*Acknowledgements.* We thank Brooke Medley for her code implementation of the Herron–Langway model, as well as Thore Kausch and
Nander Wever for interesting discussions about surface density. This work was partially supported by the Belgian Research Action through Interdisciplinary Networks (BRAIN-be) from the Belgian Science Policy Office in the framework of the project "East Antarctic surface mass balance in the Anthropocene: observations and multiscale modelling (Mass2Ant)" (contract no. BR/165/A2/Mass2Ant), and Marie Cavitte is a Postdoctoral Researcher of the F.R.S-FNRS. Hugues Goosse is the research director within the F.R.S.-FNRS. and Sarah Wauthy has benefited from a FRS-FNRS Research Fellow grant. This research contributes to the AntArchitecture action group of SCAR.



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
