# Peer review of "Investigating the spatial representativeness of Antarctic ice cores: A comparison of ice core and radar-derived surface mass balance"

_The Cryosphere, 2023_

## Referee Comment (RC1)

Review of Cavitte et al., 2023: *"Investigating the spatial representativeness of Antarctic ice cores: A comparison of ice core and radar-derived surface mass balance"*, submitted to The Cryosphere.

This paper represents an extended study of the previous work of Cavitte et al. (2022), where the authors derive surface mass balance (SMB) from ground-based radio-echo sounding (RES) data at ice rises along Dronning Maud Lands' coast (East Antarctica) as well as at the Dome Fuji site. The RES SMB is compared to SMB data derived from ice cores intersecting with the RES data, and a thorough analysis of uncertainties and representativeness of the RES-derived SMB to the ice core SMB is performed. Moreover, several gridding products with different resolutions are computed based on the RES SMB data and analysed in terms of their spatial representativeness, which is particularly important when comparing measured SMB data with model outputs.

Overall, I found the paper and the results very interesting, and I believe it will make an excellent contribution to *The Cryosphere*. I think this work is highly relevant to the glaciological community, and I look forward to seeing more outcomes of this project in the future (I hope that's what is indicated in the conclusions). The methodology is very well explained, which enables many people to reproduce the results or apply the same methodology to other data. I have to admit that I got lost here and there due to the many different sites, abbreviations and results, which are expressed in "%", but that is a relatively minor issue. My conclusion is that this paper deserves publication after minor revisions and clarifications.

**Main thoughts:**

- Maybe it would be more precise to say at least "East Antarctic" ice cores in the title, because, actually all data is located in Dronning Maud Land.

- For most readers, it is probably obvious, but could you define/explain in one sentence the term "spatial representativeness" with respect to the objectives of the study (maybe in the last paragraph of the introduction).

- The names of the ice sites are sometimes fully written in the text and sometimes abbreviated. I think sticking to either way is fine if it is consistent. Personally, I would tend to use the full names in the text as I keep forgetting the abbreviations all the time.

- All the ice core sites are more or less located in East Antarctica's Dronning Maud Land. While reading the text, I wondered how your results on SMB representativeness and SMB pattern around the dome and ice rises compare to similar studies in West Antarctica (if any out there) or other regions in East Antarctica. This could be an excellent addition to the story.

- The results section starts with the methodology of the error analysis of radar and ice core SMB. Also, the errors are presented after the "regular" results. For my feeling, it would be better to place the radar and ice core SMB uncertainty methodology in the method section and to present the uncertainty results, e.g., in section 5.4. where the spatial representativeness is discussed or in a separate subsection in the results.

- Define w.e. yr$^{-1}$ where it first appears.

**Detailed points:**

**Table 1:** While reading the text, I had problems memorising the abbreviations of the sites. Maybe it helps (at least for me, it would) to add a column in Table 1 with the full site name. I am aware that they are already mentioned in the figure caption of Figure 1, but I have the feeling that it might be easier for the reader to have the Site names and abbreviations in a table.

**L97, "across-track data density":** maybe add that this refers to the profile spacing of the radar surveys.

**Figure 1:**

- For most of the readers it might be completely clear where in Antarctica inset (a) Is located. But I think another overview map of entire Antarctica highlighting the outline of (a) would make everything clear.
- What is the grey stuff in (g)?

**L113-114 (Section 4.1):** This section is about ice core SMB, and you refer to the published data in Table 1 for the key source data. In Table 1, however, it says in the caption that it shows "key references for all radar surveys in this study". Is this correct? I am a bit confused…

**L117:** Has $R^2$ been introduced?

**L119-120, "Note that applying a Herron-Langway depth-density fit (Herron and Langway, 1980), as applied for the radar-derived data gives the same R2 values (within ±0.02) of the R2 exponential fits."** What is the context of this statement, or better, what does this mean with respect to the density estimation of the ice cores?

**L160-163:** This is a very nice summary. What about moving this statement to the beginning of subsection 4.2.1 and incorporating it into the first sentence?

**L292, "[…] the very SMB uncertainties […]":** word missing between very and SMB?

**L327-L356:** This is a very long paragraph. If possible, try to make it two paragraphs that reflect two topics.

**L371-372:** Single sentence paragraph.

**L409-414, Conclusions:** The first four sentences begin with "We". If possible, please paraphrase.

**L432: "Another interesting exercise would be to [….]".** The word "exercise" is probably not optimal here.
* * *
Thanks again for the interesting read.

Steven Franke

---

## Author Comment (AC1)

**Response to RC1 on the submitted paper *From ice core to ground-penetrating radar: representativeness of SMB at three ice rises along the Princess Ragnhild Coast, East Antarctica**

*This paper represents an extended study of the previous work of Cavitte et al. (2022), where the authors derive surface mass balance (SMB) from ground-based radio-echo sounding (RES) data at ice rises along Dronning Maud Lands' coast (East Antarctica) as well as at the Dome Fuji site. The RES SMB is compared to SMB data derived from ice cores intersecting with the RES data, and a thorough analysis of uncertainties and representativeness of the RES-derived SMB to the ice core SMB is performed. Moreover, several gridding products with different resolutions are computed based on the RES SMB data and analysed in terms of their spatial representativeness, which is particularly important when comparing measured SMB data with model outputs.*

*Overall, I found the paper and the results very interesting, and I believe it will make an excellent contribution to The Cryosphere. I think this work is highly relevant to the glaciological community, and I look forward to seeing more outcomes of this project in the future (I hope that's what is indicated in the conclusions). The methodology is very well explained, which enables many people to reproduce the results or apply the same methodology to other data. I have to admit that I got lost here and there due to the many different sites, abbreviations and results, which are expressed in "%", but that is a relatively minor issue. My conclusion is that this paper deserves publication after minor revisions and clarifications.*

We thank the reviewer for his very positive review and confirm that we indeed intend more outcomes from this project! We have now responded to all the specific comments below and in the manuscript.

**Main thoughts:**

*Maybe it would be more precise to say at least "East Antarctic" ice cores in the title, because, actually all data is located in Dronning Maud Land.*

This has now been added. Including also the suggestions of the other reviewer, we have modified it to *Investigating the spatial representativeness of East Antarctic ice cores: A comparison of ice core and radar-derived surface mass balance over coastal ice rises and Dome Fuji.*

*For most readers, it is probably obvious, but could you define/explain in one sentence the term "spatial representativeness" with respect to the objectives of the study (maybe in the last paragraph of the introduction).*

We have integrated this in the last paragraph of the introduction, as suggested, by adding within the paragraph:

*[...] Third, we grid the radar-derived SMB to look at the systematic differences in mean and temporal variability of the two SMB records as a function of grid cell size. **The smaller the difference, the higher the spatial representativeness of the point measurements is. In other words, the spatial representativeness is how much a local measurement differs from an average over a larger surface and how this difference changes as a function of the surface considered. Spatial representativeness includes all processes that introduces SMB changes over otherwise homogeneous terrain (depositional, post-depositional and orographic) as they cannot be disentangled from the method we outline.** The ultimate goal of this study is to discuss the implications of these differences for interpreting point measurements of SMB. [...].*

*The names of the ice sites are sometimes fully written in the text and sometimes abbreviated. I think sticking to either way is fine if it is consistent. Personally, I would tend to use the full names in the text as I keep forgetting the abbreviations all the time.*
We have now written out the name in full of each ice rise in the text.

*All the ice core sites are more or less located in East Antarctica's Dronning Maud Land. While reading the text, I wondered how your results on SMB representativeness and SMB pattern around the dome and ice rises compare to similar studies in West Antarctica (if any out there) or other regions in East Antarctica. This could be an excellent addition to the story.*
We are aware of several studies that attempt to reconstruct SMB by integrating ice cores and reanalysis data or model output, using spatial coherence patterns (e.g. Medley et al., 2019, *Nature Climate Change*) or data assimilation (e.g. Dalaiden et al., 2021, *Climate Dynamics*) but we are not aware of any study that quantifies the spatial representativeness of ice cores per say in other areas of Antarctica. Following the suggestion of the other reviewer, we have extended the introduction's final paragraph to state that we qualitatively know that ice rises are sites where ice cores are not expected to be representative of a large surface area however because of their complex topography.

*The results section starts with the methodology of the error analysis of radar and ice core SMB. Also, the errors are presented after the "regular" results. For my feeling, it would be better to place the radar and ice core SMB uncertainty methodology in the method section and to present the uncertainty results, e.g., in section 5.4. where the spatial representativeness is discussed or in a separate subsection in the results.*
We have followed this suggestion and have moved the entire Error Analysis section into the Methods section.

*Define w.e. yr$^{-1}$ where it first appears.*
This has been done.

**Detailed points:**

*Table 1: While reading the text, I had problems memorising the abbreviations of the sites. Maybe it helps (at least for me, it would) to add a column in Table 1 with the full site name. I am aware that they are already mentioned in the figure caption of Figure 1, but I have the feeling that it might be easier for the reader to have the Site names and abbreviations in a table.*
The full names of the sites has been added in Table 1.

*L97, "across-track data density": maybe add that this refers to the profile spacing of the radar surveys.*
This has been added as *while across-track data density (profile spacing of the radar surveys) varies a lot per survey design and site*

*Figure 1:*
*-For most of the readers it might be completely clear where in Antarctica inset (a) is located. But I think another overview map of entire Antarctica highlighting the outline of (a) would make everything clear.*
*-What is the grey stuff in (g)?*
-An inset of Antarctica has been added to panel (a).
-The grey patch was a no-data area that was present in REMA v1. We have now updated Fig. 1 with REMA v2 which is complete over this region.

*L113-114 (Section 4.1): This section is about ice core SMB, and you refer to the published data in Table 1 for the key source data. In Table 1, however, it says in the caption that it shows "key references for all radar surveys in this study". Is this correct? I am a bit confused...*

Table 1 does contain the key references for the ice core and the radar data, but this was not clear from the table title. We have now changed the title to *Radar system characteristics and key references for the radar surveys and ice core data used for each site in this study.*

*L117: Has $R^2$ been introduced?*

We now add that $R^2$ is the coefficient of determination, and point the reader to see Supplementary material S1, where we have added the following detailed description:

*$R^2$ is defined as:*

$$R^2 = 1 - \frac{SS_{\mathrm{res}}}{SS_{\mathrm{tot}}} \tag{1}$$

*where $SS_{res}$ is the sum of squares of residuals between the raw density values and the best-fit values and $SS_{tot}$ is the total sum of squares (i.e. a measure of the variance of the data).*

*L119-120, "Note that applying a Herron-Langway depth-density fit (Herron and Langway, 1980), as applied for the radar-derived data gives the same R2 values (within ±0.02) of the R2 exponential fits." What is the context of this statement, or better, what does this mean with respect to the density estimation of the ice cores?*

By this, we wanted to show that the exponential fit of the densities at the ice core site and the Herron-Langway fit of the densities at each radar point are comparable. We have added the following sentence just after: *This implies that the ice core density exponential fits and radar-derived Herron-Langway density fits are relatively similar.*

*L160-163: This is a very nice summary. What about moving this statement to the beginning of subsection 4.2.1 and incorporating it into the first sentence?*

Thank you. We have followed your advice and have regrouped that paragraph with the first sentence, to become:

*In summary, we use the published IRH twtts and the co-located ice core chronologies, to obtain best-fit density profiles, allowing us to calculate a SMB record at each radar data point. This ensures that the SMB rates are calculated from the radar data consistently across all the sites studied. Note that this is why the ages of the IRHs and the SMB history reconstructed from the radar differs slightly from previously published studies. We refer readers to Cavitte et al (2022) for more details about our method. Table 2 provides a summary of the radar IRH data sets used.*

*L292, "[...] the very SMB uncertainties [...]": word missing between very and SMB?*

It should have read *the very large SMB uncertainties relative to the absolute SMB rates.*

*L327-L356: This is a very long paragraph. If possible, try to make it two paragraphs that reflect two topics.*

We have split it into two parts, the new paragraph starting with *The difference in mean SMB increases beyond 10 % of the ice core mean SMB for a surface area that varies between a minimum of ~0.4 km$^2$ at KM and a maximum of 9.6 km$^2$ at Leningrakollen.*

*L371-372: Single sentence paragraph.*

We have regrouped this sentence with the previous paragraph.

*L409-414, Conclusions: The first four sentences begin with "We". If possible, please paraphrase. Single sentence paragraph.*

We have modified these four sentences to minimize the use of *we* at the start of each sentence, please see marked updated manuscript.

*L432: "Another interesting exercise would be to [. . . ]". The word "exercise" is probably not optimal here.*

We have reworded it to *It would also be interesting to use the radar-derived regional means [...]* .

---

## Author Comment (AC2)

**Response to R2 on the submitted paper *From ice core to ground-penetrating radar: representativeness of SMB at three ice rises along the Princess Ragnhild Coast, East Antarctica**

*The paper deals with an interesting and important topic, representativeness of point SMB observations in Antarctica. But it mainly uses data of ice rises, which are known to have a significant impact on precipitation amounts. This means that the context is different from what the reader could derive from the title, and the title and text should be changed to reflect this. See more detail in Major Comments below. The paper is also overly long and sometimes difficult to read, with rather long-winding sentences, and I invite the authors to attempt to write more concisely and shorten the paper. The figures are of good quality.*

We thank the reviewer for their constructive review. We have addressed all comments here below and in the manuscript. To answers a few of the specific points raised here:

- We have modified the title in light of your and the other reviewer's suggestions to *Investigating the spatial representativeness of East Antarctic ice cores: A comparison of ice core and radar-derived surface mass balance over coastal ice rises and Dome Fuji*, so that readers are clear on the focus of our study.

- We agree that the paper is rather long but we wanted our work to be clear so that the same methodology could be applied elsewhere. We prefer to keep all the elements in the main manuscript rather than have a long supplement.

- We have gone through the manuscript and split long sentences for clarity (see marked-up copy).

**Major comments**

*My main concern is the context and framing of this study. Upon first sight, the study is positioned as quantifying the representativeness of point SMB measurements. This implies that you are mainly interested in the impact of post-depositional processes (sublimation, drifting snow etc.) that introduce SMB differences in otherwise homogeneous terrain. But this is not so. The work compares ice core-derived SMB from the summit of ice rises to radar-derived SMB measurements in the vicinity and intersecting the summit. Ice rises cannot be considered homogeneous terrain. Several studies (e.g.: doi:10.3189/2014JoG14J040) show that ice rises cause important depositional (precipitation) SMB gradients, which are qualitatively well understood and modelled. In other words, nobody would normally claim that an SMB record of the ice rise dome is representative for its surroundings. And this paper shows this is indeed the case. This is different for Dome F, where the terrain is smooth, accumulation amounts are much lower, and the topographic precipitation effect is expected to be small. This is confirmed by the much smaller regional differences there. So the authors should include a discussion in which they motivate their choice for using ice rises to address the issue of SMB representativity, and how depositional and post-depositional signals can be disentangled.*

We state explicitly in the last paragraph of the introduction that *The nine sites studied in this work were chosen because of their data availability and the grid-like design of the surveys which sample SMB in all directions over varied surface topography more homogeneously*. We have now added in the same paragraph that we *expect* ice rise to not be very representative of SMB spatially due to these SMB depositional gradients, as follows:

*These eight sites can thus be easily compared.* **These eight sites can thus be easily compared. We do highlight that because of their topography, ice rises cause significant spatial variability of the SMB that is qualitatively well understood and modelled (Lenaerts et al., 2014). Ice core records from ice rises are therefore expected to be representative of a small surface area but it is important to quantify how small. Because ice rises are coastal with a simple ice flow regime and high accumulation, they have been strategic drilling sites that produce high resolution records. Consequently, ice rises will certainly continue to be drilled in the coming future and it is highly relevant to quantify their regional representativeness.** *The ninth site is the Dome Fuji plateau region which has a low accumulation rate and very gentle surface slopes as opposed to the coastal sites,* **where the orographic precipitation effect is expected to be small**.

Finally, we have added a sentence to state explicitly that depositional and post-depositional processes cannot be disentangled using our method:

*Note that depositional and post-depositional processed both reduce the spatial representativeness of the SMB signal but these cannot be disentangled from the method we outline.*

*l. 18: "However, these data clearly indicate that local spatial SMB variability must be considered when assessing mass balance as well as comparing modeled SMB values to point field data." Yes, but how must they be considered? By assigning an uncertainty to the observation?*

Yes, that is what we suggest, and we now make it explicit in the abstract: *However, these data clearly indicate that local spatial SMB variability must be considered when assessing mass balance as well as comparing modeled SMB values to point field data* **and must therefore be included in the estimate of the uncertainty of the observations**.

*l. 23: "rising specific humidity" It is not specific humidity, but the saturation specific humidity that will increase.*

Modified.

*l. 29: "The Antarctic coastal region is getting most of the accumulation due to source proximity and low elevation (e.g., Lenaerts et al., 2019)" Yes, but the main reason is topographic precipitation forcing, which also explains why local accumulation rates can be higher over the lower grounded ice sheet compared to the neighbouring flat ice shelves.*

We agree and have modified our sentence to reflect this: *The Antarctic coastal region is getting most of the accumulation due to a combination of source proximity, low elevation and topographic precipitation forcing [...]*

*Minor comments*

*l. 1: over −> of*
Done.

*l. 3: validate −> evaluate (this is about models, which cannot be validated by default)*
We have used the word *assess* instead to avoid redundancy.

*l .8: "between annual and sub-decadal resolution": annual is also decadal, please make more concrete.*
We have changed it to *yearly and multi-year*.

*l. 14: areas −> area*

Done.

We have reformulated to: *And averaging many records in close proximity does not improve signal-to-noise in the ice core records [...].*

The sentence now reads: *In this study, we compile various radar-derived SMB datasets and describe their spatial variability referenced to co-located ice core SMB.*

We have changed it to: *where radar stratigraphy has a vertical resolution of a few years*.

We have changed it to *and very gentle surface slopes*.

We have rounded all uncertainties to a single decimal.